# Three-dimensional skyrmionic cocoons in magnetic multilayers

Matthieu Grelier [1], Florian Godel [1], Aymeric Vecchiola [1], Sophie Collin[1], Karim Bouzehouane [1], Albert Fert [1], Vincent Cros [1] & Nicolas Reyren [1]

Three-dimensional spin textures emerge as promising quasi-particles for encoding information in future spintronic devices. The third dimension provides more malleability regarding their properties and more flexibility for potential applications. However, the stabilization and characterization of such quasi-particles in easily implementable systems remain a work in progress. Here we observe a three-dimensional magnetic texture that sits in the interior of magnetic thin films aperiodic multilayers and possesses a characteristic ellipsoidal shape. Interestingly, these objects that we call skyrmionic cocoons can coexist with more standard tubular skyrmions going through all the multilayer as evidenced by the existence of two very different contrasts in room temperature magnetic force microscopy. The presence of these novel skyrmionic textures as well as the understanding of their layer resolved chiral and topological properties have been investigated by micromagnetic simulations. Finally, we show that the skyrmionic cocoons can be electrically detected using magneto-transport measurements.

Topological magnetic textures have been under close scrutiny in recent years as they could represent an original asset for the development of new logic and memory devices as well as novel hardware neuromorphic nanocomponents[1]. Chiral magnets and magnetic multilayers allow for the stabilization of two-dimensional (2D) textures that have been extensively investigated[2]. A key example remains the magnetic skyrmion[3], a 2D whirling of the magnetization which is therefore uniform along the vertical dimension (the dimension normal to the film), first observed in B20 chiral magnets[4]. In thicker structures, skyrmion tubes can also be stabilized, thus adding a vertical dimension that presents small variations depending on the equilibrium of the magnetic interactions at play[5].

More recently, however, interest has surged for three-dimensional (3D) nanomagnetism and consequently for complex textures displaying an important evolution over the thickness[6–8] leading to the discovery of new categories of topological textures. For instance, in bulk material, theoretical predictions, followed by experimental observation, describe magnetic bobbers[9,10] where one end of a skyrmion string goes to a surface while the other one ends with a point singularity inside the sample. Another case of these skyrmions strings

corresponds to torons[11,12], otherwise known as dipole strings due to the two Bloch points at their extremities. Analogous textures have been reported as well in magnetic multilayers with the recent observation of the elusive hopfion[13,14], originally predicted to appear in chiral magnetic crystals[15,16]. It was also shown that, in hybrid ferromagnetic/ferrimagnetic multilayers, truncated skyrmions that end abruptly at a given interface can be stabilized[17,18].

Such new objects pave the way toward expending spintronics into the third dimension but several challenges must be overcome first, mostly their characterization as well as their implementation. Regarding the former, studying such textures require to probe the magnetization over the thickness which is often achieved by using recent high-end imaging techniques that rely on the use of soft X-rays or electrons. For example, remarkable reconstructions have been achieved for a permalloy disk using X-ray laminography[19] or for various textures with X-ray tomography[20–23].

Here, as a less challenging and restrictive alternative, we put forward a simpler approach based on magneto-transport measurements that can be accurately coupled with micromagnetic simulations not only to detect 3D objects but to resolve their magnetization profile as

[1]Unité Mixte de Physique, CNRS, Thales, Université Paris-Saclay, 91767 Palaiseau, France. e-mail: nicolas.reyren@cnrs-thales.fr

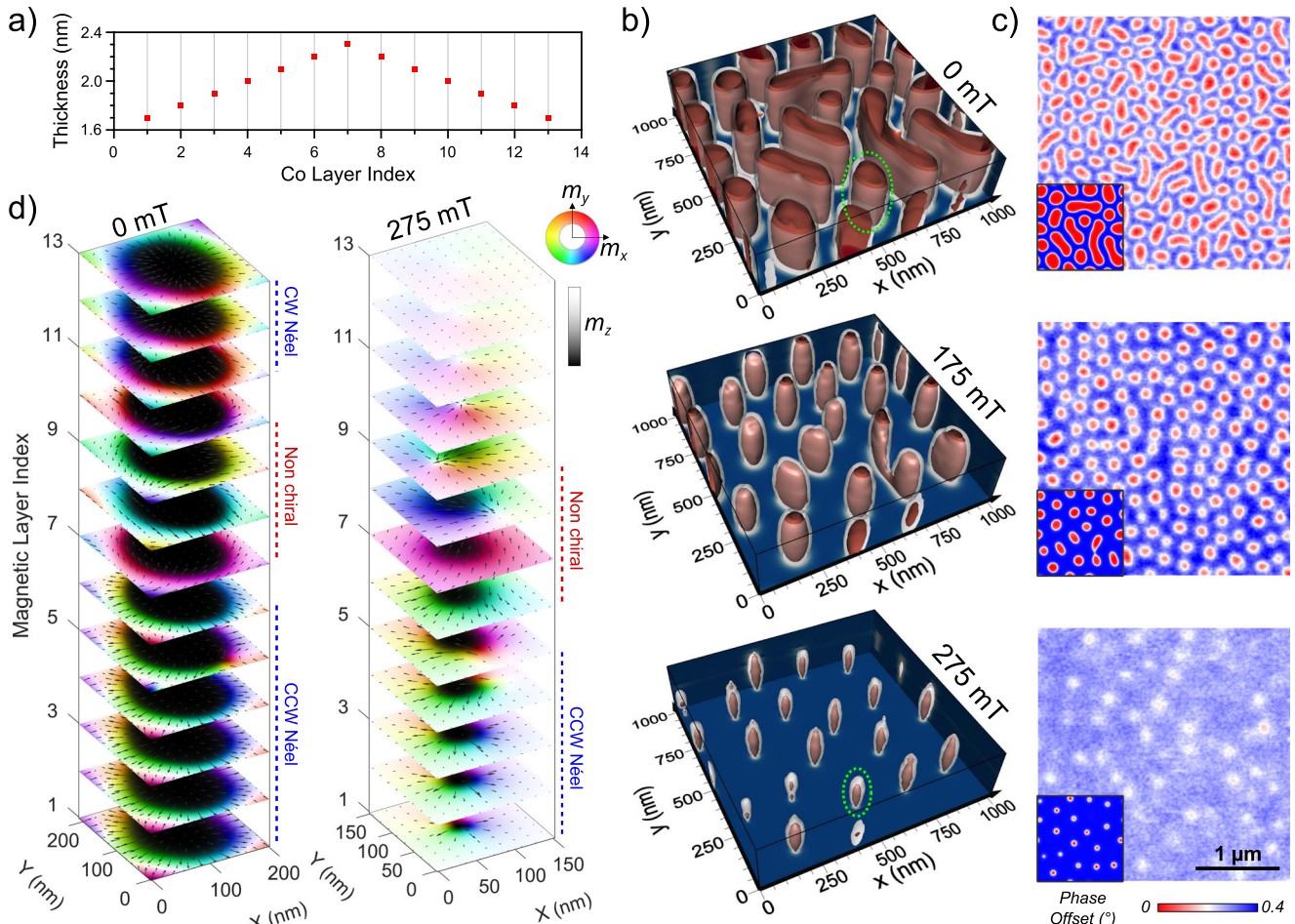

**Fig. 1 | Properties of Single Gradient (SG) multilayers with parameters $S = 0.1$ nm, $X_1 = 1.7$ nm, $N = 13$ layers. a** Evolution of the Co thickness for a typical SG multilayer, defined by the number of layers $N$, the thickness step $S$, and the starting thickness $X_1$. **b** Field evolution of the magnetic textures with micromagnetic simulations after an out-of-plane saturation displayed with isosurfaces (in red, $m_z = -0.8$, in white, $m_z = 0$ and in dark blue, $m_z = 1$). **c** Corresponding phase map

images obtained with MFM accompanied by simulated ones ($1 \times 1 \mu m^2$) at the bottom left corner. The experimental magnetic fields correspond to the simulated ones $\pm 10$ mT and point out-of-plane. The scales shown in the bottom image are common to all MFM maps. **d**) Magnetization cuts of a selected object, indicated in (**b**) by the green dotted ellipses, at two different magnetic fields to evidence the vertical evolution and the chirality (CW: clockwise, CCW: counterclockwise).

well. Concerning their implementation, most of those newly observed textures have been detected in chiral magnetic crystals or in nanostructures which strongly limits their functionality. In opposition, multilayers appear as a promising alternative to harbor 3D topological textures as they are more resilient and scalable while offering a broader tunability of their magnetic properties. Indeed, tuning the films thickness and the interfaces properties helps to control various magnetic interactions, notably the Dzyaloshinskii-Moriya interaction (DMI)[24-26], an asymmetric exchange facilitating the stabilization of topological non-collinear textures[27,28]. Based on this strong malleability, we have developed original multilayers architectures that allow the stabilization of a new type of 3D spin texture, the skyrmionic cocoon, which could ultimately serve as a basis for 3D skyrmionic devices.

## Results

### Engineering of the multilayer properties

In order to induce the stabilization of 3D magnetic textures, we exploit the tunability of the magnetic multilayers by engineering an elaborated sample architecture (Fig. 1a) with a variable thickness for the ferromagnetic material to introduce an uneven distribution of the magnetic interactions over the vertical dimension. Typically, the multilayers are made of repetitions of the trilayers Pt 3 nm/Co $X_i$/Al 1.4 nm, with $X_i$ the Co thickness of the $i$th magnetic layer. This trilayer

system has been chosen as we recently showed that it maximizes the interfacial DMI[25], which usually facilitate the stabilization of chiral spin textures. This first sample heterostructure, that we call single gradient (SG), is designed with a specific evolution for the ferromagnetic thickness: it transitions from the initial value $X_1$ to $X_{max}$ with a fixed thickness step and then come back to the initial thickness, following the same steps in reverse order. In other words, the Co thickness $X_i$ can be recursively defined as:

$$X_{i+1} = \begin{cases} X_i + S & \text{for } i \leq N/2 \\ X_i - S & \text{for } i > N/2 \end{cases}, \quad (1)$$

with $S$ the thickness step between two consecutive layers, $N$ the total number of layers and $X_1$ the bottom layer thickness. In Fig. 1a, an example of the thickness evolution as a function of the layer index $i$ is shown for a case where the outer layers are the thinnest ones ($S > 0$). By tuning $S$ and the external perpendicular magnetic field, the vertical confinement of the magnetic textures can be controlled (Supplementary Section 1 and Fig. S1). In the following, we focus on the structure parameters used in Fig. 1, namely $X_1 = 1.7$ nm, $S = 0.1$ nm, and $N = 13$ layers, that correspond to the sample which has been experimentally characterized and studied in detail with micromagnetic simulations (see "Methods" for detailed structure). Since the reorientation transition thickness was measured to be 1.7 nm in our Pt/Co/

Al trilayers, the multilayer under consideration should mostly display an in-plane (IP) effective anisotropy. The effective anisotropy can be well modeled considering only an interfacial uniaxial axis characterized by $K_{u,s} = 1.58 \pm 0.08$ mJ/m$^2$ (see Fig. S2 for magnetization curves).

In order to apprehend the actual shape and magnetization distribution of the objects hosted in SG structures, we perform micromagnetic simulations using Mumax3[29]. To help the visualization of the 3D objects, their output is displayed making use of the isosurfaces $m_z = -0.8$ in red, $m_z = 0$ in white, and $m_z = 1$ in dark blue (only shown in the background planes). In Fig. 1b, the relaxed states after an out-of-plane (OOP) initialization are shown for three different OOP magnetic fields, $H_\perp$. At zero field, considering a planar cut, an irregular lattice of circular objects and more elongated ones is found. Both kinds are columnar, i.e., they are extending from the bottom to the top interface. Increasing $\mu_0 H_\perp$ up to 175 mT reduces their size and homogenizes their shape due to the Zeeman effect gaining in strength, resulting in a configuration more similar to a lattice of skyrmion tubes. Then at 275 mT, they become vertically confined and remain in a fraction of the layers only, acquiring a characteristic deformed ellipsoid shape. Based on this vertical confinement and their peculiar profile, we name them skyrmionic cocoons.

To experimentally characterize those new 3D magnetic textures, we first use magnetic force microscopy (MFM) measurements. In Fig. 1c, experimental phase maps are shown alongside with the simulated MFM images in the insets obtained from the previously described micromagnetic simulations. The agreement between the simulations at the three chosen fields and the actual magnetic configuration is satisfying: at zero field, a distribution of inhomogeneous skyrmionic tubes is measured that then acquire a barrel shape in the vertical direction as $\mu_0 H_\perp$ reaches 175 mT. Then for even higher fields e.g., 275 mT, in such SG, we clearly observe a decrease in MFM phase contrast coming from the fact that the detected objects are more and more buried under the top surface. This behavior is in qualitative agreement with the corresponding micromagnetic simulations which thus appear to appropriately model the density, the shape and the size of the observed objects.

In Fig. 1d, we present layer-resolved magnetization profiles simulated at 0 and 275 mT. These profiles clearly evidence the evolution from a fully columnar skyrmion to a vertically confined skyrmionic cocoon. Moreover, these simulations allow to discriminate the effective chirality of the spin textures in each magnetic layers composing the multilayers. For the case of skyrmion tubes at zero field, we find that the bottom layers (1 to 6) are counterclockwise (CCW) Néel type whereas in their top layers (11 to 13) they are clockwise (CW) Néel. This distribution is the result of an equilibrium between the interfacial DMI and the dipolar field: for the chosen stacking sequence, the CCW chirality is favored by the interfacial DMI, while the dipolar field favors CW chirality in the top layers and CCW in the bottom ones. In the remaining middle layers, the skyrmion walls display a magnetization pointing uniformly in the same in-plane direction as the interlayer dipolar interaction and the DMI nearly cancel each other out. The resulting local textures change significantly for $\mu_0 H_\perp = 275$ mT: the skyrmion tube evolves into a cocoon as the core shrinks and then disappear in the outer layers. Indeed, in the top part (layers 11 to 13), the magnetization is pointing up with a slight tilt ($\langle m_z \rangle > 0.99$ for the cocoon presented in Fig. 1d) toward the cocoon's core that is induced by the dipolar field. Directly above the cocoon, in the layers 9 and 10, this inclination is stronger ($\langle m_z \rangle > 0.85$) which yields a vortex-like texture with no singularity as the spins rotate continuously keeping $m_z > 0$ everywhere. Note the absence of Bloch points at the extremities of the cocoons, due to the discontinuous distribution of the magnetic materials in the multilayer along the $z$ direction, the discrete layers being coupled by dipolar fields only. Therefore, it differentiates them from similarly shaped textures like the aforementioned torons.

The micromagnetic simulations also allow the topological properties of skyrmionic cocoons to be investigated. However, due to their discretization along the vertical direction, an exact formulation of an appropriate 3D winding number remains problematic. As a first approach, we compute the customary 2D skyrmion number[3] in each layer which yields non-zero values for skyrmion walls with a well-defined Néel chirality, either CW or CCW. On the contrary, in the middle layers (their precise position depends on the magnetic field, see Supplementary Fig. S9) it often cancels as the skyrmion walls point uniformly in a given direction. Alternatively, we can consider the 3D generalization of the topological charge (as defined in ref. 30) for which the integral is performed on a closed surface enclosing the magnetic object, heeding that this definition is ill-defined as we study a non-continuous medium. In the Supplementary Information (Section 5), we detail how the calculated topological charge would depend on the extension and position of the cocoon in case the magnetization can be considered continuous.

## Coexistence of skyrmionic cocoons and tubes

Based on the properties of skyrmionic cocoons identified in SG multilayer, we decide to push further the engineering and the complexity of the multilayers to investigate how these new 3D magnetic objects can be combined with other textures. Beyond the fundamental interest, for example to study the transition processes between topologically different magnetic textures, the purpose here is also to investigate how these cocoons can be intentionally localized at different z-positions, that in the perspective of applications, would be a first step in the design of 3D skyrmionic platforms. The chosen multilayer structure presented in Fig. 2a is called Double Gradient (DG), as it includes 2 SG blocks separated by a number $M$ of trilayers with thin Co of constant thickness $Y$ that is chosen to have a strong perpendicular magnetic anisotropy (PMA). In the following, we focus on the architecture defined by $X_1 = 2.0$ nm, $S = 0.1$ nm, $Y = 1.0$ nm, $N = 13$ layers, and $M = 15$ repetitions.

In Fig. 2b, we present the MFM images recorded at remanence after different initialization processes and the corresponding simulated MFM images. First, saturating the sample magnetization in-plane (IP) results in stripes with various branching giving a single MFM contrast. This indicates that the textures are columnar and thus go through all the layers, as confirmed by the 3D micromagnetic simulations (see Supplementary Fig. S5). In case of an initialization with an OOP field, two different contrasts are clearly visible (white and gray) in the MFM. The related simulations told us that such a case is representative of the presence of two different objects. Note the size of the smallest features that can reach sub-100 nm size at zero field. Finally, the case of a tilted field during the initialization leads to a configuration containing distorted stripes with dots of lower contrasts in their midst.

In Fig. 3a, we present the MFM images corresponding to the tilted field evolution. As shown before, at $\mu_0 H = 0$ mT, we can distinguish two kinds of textures: first the ones that penetrate through all the layers, i.e., the columnar stripes, and second the skyrmionic cocoons. For simplicity, we will refer to the former as 3D stripes (or worms) in the following. The strong contrasts therefore correspond to the columnar objects whereas the cocoons are associated with a weaker signal. The corresponding micromagnetic simulations (see Fig. 3b) indicate that in most cases, two cocoons are present, one in each SG block, and that they sit one on top of the other. For a larger OOP magnetic field (230 mT), the red 3D stripes become narrower and the density of cocoons increases, yielding a configuration showing alternatively stripe domains and chain of cocoons. For a slightly higher field, at 350 mT (resp. 275 mT) in the simulations (resp. experiments), the elongated side of the 3D stripes shrinks, causing them to transition into skyrmion tubes (large red dots) whereas the cocoons (white small dots) appear more resilient to the field. Finally, when the field is high

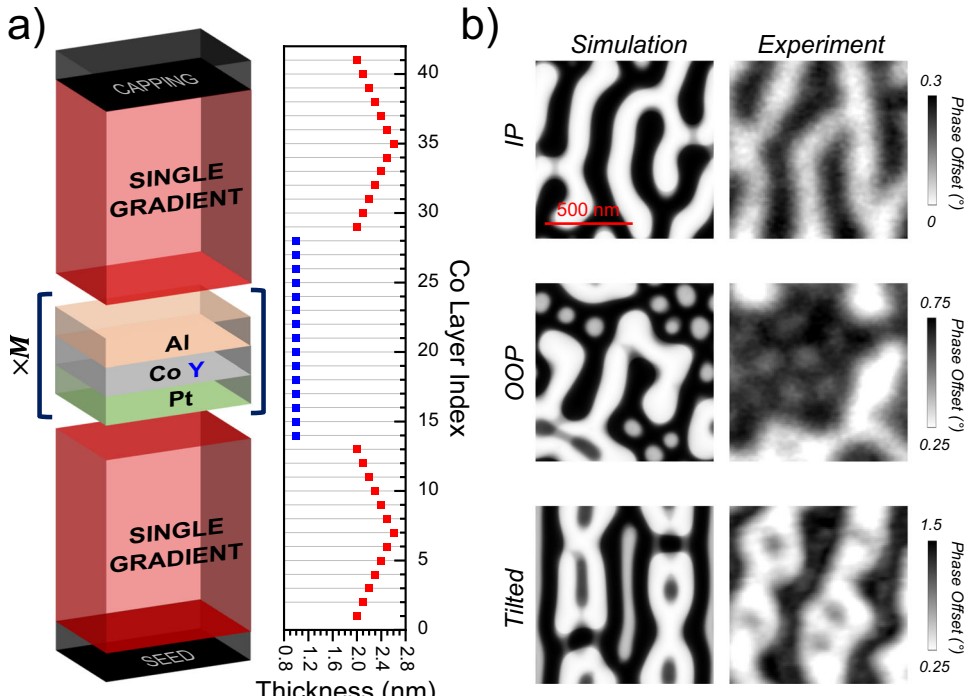

**Fig. 2 | Remanent magnetic configuration in DG multilayer with parameters $X_1$ = 2.0 nm, $S$ = 0.1 nm, $Y$ = 1.0 nm, $N$ = 13 layers, $M$ = 15 repetitions. a** Schematic structure with additional parameters of the thin Co thickness $Y$ and the associated number of repetitions $M$. The lateral plot shows the Co thickness evolution. **b** Simulated and experimental phase maps for different magnetic histories. For the simulations (resp. the measurements), the sample was either saturated (resp. demagnetized) out-of-plane (OOP), in-plane (IP) or with a tilted magnetic field, 30° away from the normal (resp. 60°). All images share the same length scale, displayed in the top left image.

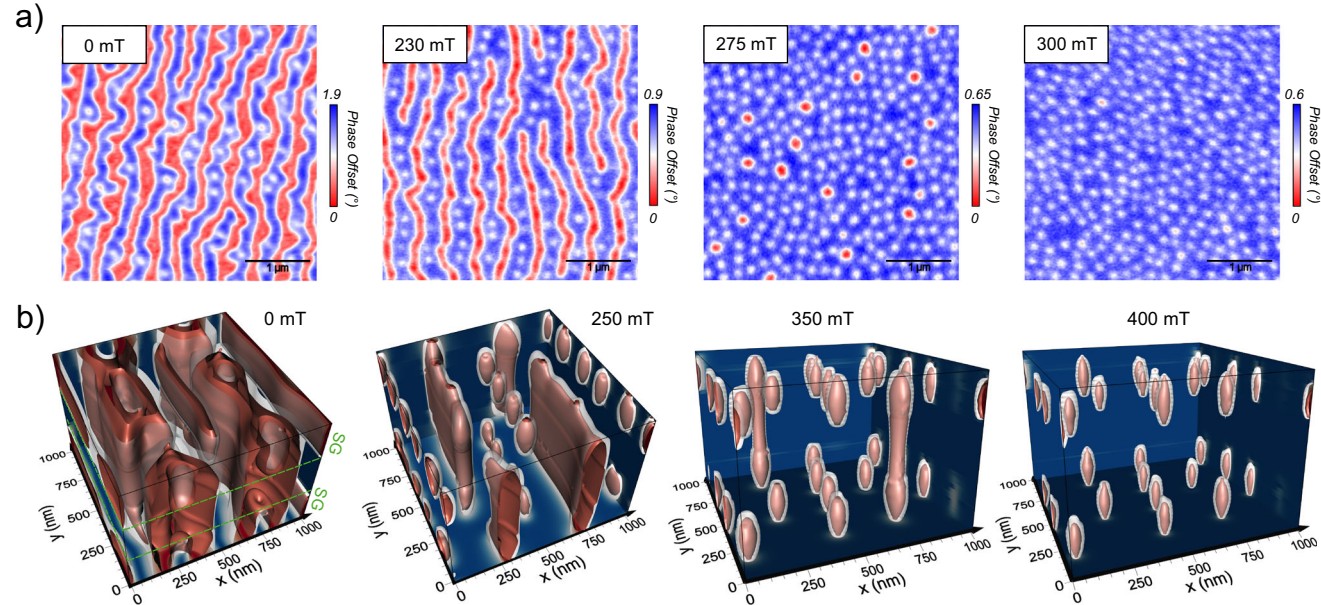

**Fig. 3 | DG multilayer field dependency. a** Experimental MFM phase maps measured after a 30° demagnetization (away from the normal). **b** Relaxed states of micromagnetic simulations, displayed with isosurfaces (in red, $m_z = -0.8$, in white, $m_z = 0$ and in dark blue, $m_z = 1$), at various magnetic fields starting from a 60° initialization. The magnetic field is applied perpendicularly to the sample.

enough, at 400 mT, the layers with strong PMA point uniformly in its direction, thus transforming the remaining skyrmion tubes into two cocoons, one in each gradient. We hence identify a field range in which we can stabilize cocoons in separated blocks which could be piled up in more complex architectures. The exact nature of the transitions at play and their associated mechanisms remain to be elucidated. Finally, note that in the simulations, most of the cocoons appear to be vertically aligned with another. As the MFM signal comes mainly from the stray field of the top layers, this cannot be yet confirmed experimentally. To do so, more advanced 3D imaging techniques such as

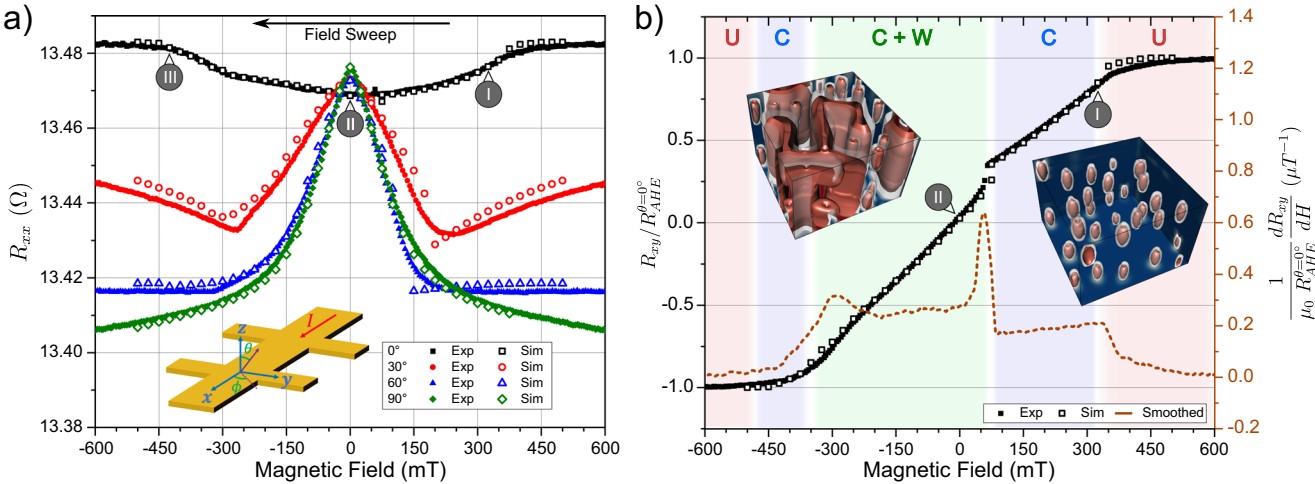

**Fig. 4 | Electronic transport measurements on $20 \times 100\,\mu m^2$ Hall bars for a DG, associated with the corresponding micromagnetic simulations.**
**a** Measurements of $R_{xx}(H)$ for different inclinations of the magnetic field. The inset shows the geometry of the experiment. **b** $R_{xy}(H)$ at $\theta = 0°$ (OOP field) and its derivative. The background colors corresponds to the magnetic phases (U: uniform, C: cocoons, W: worms) as predicted by the simulations. The displayed simulations are associated with the points indicated by a roman number and represent a $1 \times 1\,\mu m^2$ area. The experimental $R_{xy}$ has been normalized by $R_{AHE}^{\theta=0°}$ to correspond to the associated $m_z$ value. All the displayed curves, measured or simulated, are acquired from positive field saturation, sweeping the field towards negative values in the $yz$ plane.

laminography would be necessary in order to resolve the precise distribution of the magnetization.

## Measuring the electrical signature of cocoons

To investigate the presence of cocoons in the bottom layers, in addition to the MFM imaging, we perform magneto-transport measurements in Hall bar devices of size $20 \times 100\,\mu m^2$ in the previously studied DG multilayer to probe their electrical properties. We measure both the longitudinal resistance $R_{xx}$ and the transverse resistance $R_{xy}$ while sweeping the magnetic field along different orientations. In order to fit these electrical measurements, a simple model taking into account several possible contributions to the magneto-resistive effects is considered[31]. The current is injected along $x$, the film normal is along $z$ and $\theta$ corresponds to the angle between $z$ and the field (see inset of Fig. 4a). First, due to the presence of heavy metallic layers in contact with the magnetic films, the spin Hall effect (SHE) in the heavy metal gives rise to the spin Hall magnetoresistance (SMR)[32], that depends on the angle between the magnetization direction **m** of the ferromagnet and the spin polarization due to the SHE[33,34]. The change in $R_{xx}$ due to the corresponding SMR is thus proportional to $m_y^2$. The second contribution is the anisotropic magnetoresistance (AMR)[35], that is proportional to $m_x^2$, i.e., when the magnetization is parallel to the current. Finally, the evolution of the transverse resistance is governed by the anomalous Hall effect (AHE)[32,36], that is proportional to the out-of-plane component of the magnetization: $R_{xy} \propto m_z$. From those considerations, $R_{xx}$ and $R_{xy}$ can be written as:

$$R_{xx} = R_{xx}^0 + R_{SMR}\,m_y^2 + R_{AMR}\,m_x^2, \qquad (2)$$

$$R_{xy} = R_{xy}^0 + R_{AHE}\,m_z, \qquad (3)$$

With $R_{xx}^0$ the longitudinal resistance for $\mathbf{m} = m_z\hat{z}$ and $R_{xy}^0$ is an offset in the transverse resistance of the multilayer. The coefficients $R_{AMR}$, $R_{SMR}$ and $R_{AHE}$ are the proportionality coefficients of the associated effects. In the DG multilayer, the first two are found to be negative (see "Methods"). Note that the contributions related to ordinary Hall effect, giant magnetoresistance (GMR) and the planar Hall effect (PHE) are negligible and thus not considered. Moreover, as we found in previous work[37] that the topological Hall effect (THE) contribution in standard skyrmion multilayers was negligible compared to AHE, we do not include this contribution either. In Fig. 4, we display the measured resistance as a function of the applied field alongside the predicted evolution of $R_{xx}(H)$ using the $m_x$, $m_y$, and $m_z$ components from the micromagnetic simulations as inputs in the previous equations. Note that the coefficients of Eqs. (2) and (3), $R_{AMR}$, $R_{SMR}$ and $R_{AHE}$, are fixed by experimental measurements, and therefore, there are no fitting parameters for the curves in Fig. 4.

In the continuity of the previous measurements, we first consider $R_{xx}$ with an OOP field ($\theta = 0°$). Only minor variations are expected since the in-plane components should remain small throughout the field sweep. Starting from high positive field to negative field, the magnetization is first saturated along the $z$ direction until the first skyrmionic cocoons nucleate in both SG near 325 mT (state I). Their apparition create in-plane components which yield, due to the sign of the AMR and SMR, a negative contribution to $R_{xx}$. As they expend in all directions upon lowering $\mu_0 H$, it continues to decrease, reaching its minimum at zero field (state II). According to the simulations, it corresponds to a state in which 3D worms coexist with the skyrmionic cocoons. The symmetric trend is then observed for negative magnetic field and it is similar to the one previously described with the MFM study: the spin textures shrink and the 3D worms end up disappearing, leaving only cocoons (state III) until the sample saturates.

To further validate our approach, the field was tilted at different angles (30, 60, 90°) in order to introduce more in-plane components. This leads to a stronger field response that challenges the sensitivity of the micromagnetic simulations. Since the magnetic field lies partially (or totally) in-plane, the nucleation events provoke significant positive jumps in the resistance linked to a loss of IP components. It is also noticeably harder to saturate the sample along those directions due to the strong PMA layers, leaving variations at higher fields. The application of a tilted magnetic field leads to complex spin textures that are described in Supplementary (Fig. S5). At zero field, however, we retrieve a state with 3D stripes and skyrmionic cocoons for $\theta = 30, 60°$ whereas $\theta = 90°$ yields only 3D stripes (Fig. 2). Regardless of the angle of the field, the resistances extracted from the simulations fit remarkably well the experimental ones. The difference between the two is typically of a few m$\Omega$ which can originate from the simplicity of our model and its neglected contributions as well as from some experimental angular inaccuracy (<0.5°).

In Fig. 4b, $R_{xy}(H)$ for $\theta = 0°$ is displayed, along with its experimental derivative with a colored background associated with the different magnetic phases: uniform (red), cocoons (blue), and cocoons alongside with 3D worms (green). Based on the simulations and those curves, it is possible to identify more precisely the different magnetic phases and transitions that the sample undergoes. Near 75 mT, we can notice a jump in the resistance: it corresponds to the nucleation inside the strong PMA layers which was negligible in $R_{xx}$ because the associated domain walls are quite narrow. Therefore, above that field, the DG multilayer hosts only skyrmionic cocoons in the SG blocks as illustrated with the display of state I. The field range of the associated state can be pinpointed looking at the derivative where a wide plateau is visible, before the sharp peak coming from the nucleation in the middle layers. For negative magnetic field, simulations place the regime with only cocoons between −350 and −475 mT, roughly before the small peak in the derivative near −300 mT. In the end, electrical measurements allow to recognize most of the magnetic phases of the DG multilayers and, for more precision, it can be coupled with micromagnetic simulations.

## Discussion

By using variable thickness of the ferromagnetic element in magnetic multilayers, we have demonstrated the existence of skyrmionic cocoons, a new type of 3D topological magnetic texture vertically confined with sizes under 100 nm that could be seen as discretized torons. Moreover, by studying more elaborated architectures, we have shown that they can be observed conjointly with columnar textures even in the absence of an external magnetic field and that they can be located at different vertical levels of the structure. Thanks to the excellent agreement between the simulations and the electronic transport measurements, the former give us a precise insight into the three-dimensional distribution of the magnetization as well as the different magnetic phases. This achievement also promotes magnetoresistance measurements as an easily implementable approach to detect 3D objects in such magnetic multilayers. In this study, we focused on a particular set of parameters but various other architectures based on the ones presented could be envisioned to tune the skyrmionic cocoons properties. For example, playing with the thickness step can impact their confinement and by varying the number of gradient blocks in the structure, they would have more different vertical positions available. Following that reasoning, a memory device could ultimately be based on them, by encoding multiple states with the vertical position and number of cocoons for instance. Thus, this coexistence of various magnetic textures in multilayers, the discovery of those objects that we propose to call skyrmionic cocoons as well as their malleability are particularly interesting as they can open new paths for three-dimensional spintronics.

## Methods

### Sample preparation and characterization

The samples have been grown on thermally oxidized silicon substrates using magnetron sputtering at room temperature. A seed layer of 5 nm Ta is typically used and 3 nm Pt capping is deposited to protect from oxidation. In the characterized structures, the Al is fixed at 1.4 nm whereas the Pt is 3 nm in the SG and the strong PMA part of the DG but 2 nm in the gradient parts of the DG. In details, the SG corresponds to $SiO_x$|Ta5|Pt3|(Co[1.7:0.1:2.3]|Al1.4|Pt3)(Co[2.2:0.1:1.7]|Al1.4|Pt3) and the DG to $SiO_x$|Ta5|Pt3|(Co[2.0:0.1:2.5]|Al1.4|Pt2)(Co[2.6:0.1:2.0]| Al1.4|Pt2)|(Co1.0|Al1.4|Pt3) × 15|(Co[2.0:0.1:2.5]|Al1.4|Pt2)(Co[2.6:0.1: 2.1]|Al1.4|Pt2) (Co2.0|Al1.4|Pt3) where the notation $[X_1 : S : X_2]$ corresponds to the thickness sequence between $X_1$ with $S$ being the thickness step. The magnetic hysteresis have been measured using AGFM (see Supplementary Fig. S2), and SQUID to determine the saturation magnetization and the corresponding paramagnetic contribution from the substrate. SQUID measurements yield a

magnetization saturation of $1.23 \pm 0.02$ MA/m for the SG and $1.22 \pm 0.02$ MA/m for the DG.

### Magnetic force microscopy and demagnetization

Before imaging, the samples have been demagnetized using an electromagnet which applied an oscillating magnetic field of exponentially decreasing amplitude. The MFM images have been acquired using cantilevers from TeamNanotech (TN) with nominal stiffness of 3 N/m, with its tip capped by 7 nm of magnetic (proprietary) material and 10 nm of Pt. We used a tapping mode with a lift height of 10 nm, at 75% of the drive amplitude with double-passing and at room temperature. Note that the lift height is defined relatively to the first pass height recording the topography, implying that the effective height of the probe is larger than 10 nm. The phase maps displayed have been modified using a Gaussian filter as implemented in Gwyddion software[38] with a 2 pixels FWHM.

### Electronic transport measurements

The DG sample has been patterned using UV lithography into Hall bars of dimensions $20 \times 100$ µm². Then it had been measured in a field setup with an electromagnet able to reach 0.65 T. To ensure saturation of the DG regardless of the field angle, the measurements have been repeated using a second setup, able to reach fields up to 9 T, to extract the appropriate AMR and SMR values. The current applied through the Hall bridges was typically 100 µA.

The $R_{SMR}$ and $R_{AMR}$ constants of the previous equation are measured from $R_{xx}(\theta)$ measurements for an external magnetic field which is large enough to saturate the magnetization, rotating in the (xz) or (yz) plane, $\theta$ being the angle of the field with the film normal. To limit the Lorentz magnetoresistance, the magnetic field used for the measurement was slightly above the saturation field for each sample. For the DG, we find a negative contribution for the AMR and SMR effects: $R_{AMR}^{DG} = -18$ mΩ and $R_{SMR}^{DG} = -108$ mΩ.

### Micromagnetic simulations and 3D representation

The simulations have been run using the micromagnetic solver Mumax3[29] and the following micromagnetic parameters: the exchange constant $A = 18$ pJ/m, the saturation magnetization $M_S = 1.2$ MA/m, the uniaxial surfacial anisotropy constant $K_{u,s} = 1.62$ mJ/m² and the DMI surfacial constant $D_s = 2.34$ pJ/m. To appropriately represent the complex structure of the samples, cells of $2 \times 2 \times 2.1$ nm³ were used for the SG and $2 \times 2 \times 1.8$ nm³ for DG for a total simulation space of, respectively, $512 \times 512 \times 37$ and $512 \times 512 \times 121$ cells. As the domain wall can be thin in the strong PMA layers, and thus the magnetization can rotate over relatively short lengths (maximum angle below 30°), the simulations have been checked by running more precise simulations with $1 \times 1$ nm² instead of $2 \times 2$ nm² which did not yield a significant change in the magnetization distribution. Given the variable thickness, each trilayer was divided in three layers, one magnetic and the other two empty and the magnetic parameters were diluted depending on the experimental thickness of the given magnetic layer. In practice, we define the dilution factor $f = X_i/L_z$ with $L_z$ the vertical dimension of the cells. Then, the micromagnetic parameters are updated: for the exchange, the DMI and the saturation magnetization, the constants are simply multiplied by $f$ and for the anisotropy we set $K_u = K_u f - \mu_0 M_S^2 (f - f^2)/2$[39] with $K_u = K_{u,s}/X_i$.

The MFM images have been simulated using a lift height of 50 nm, using the method implemented in Mumax3.

For the electronic transport measurement simulations, the magnetization was initialized in a given direction (usually along $y$ and $z$) with 50% noise and we swept the field from positive values to negative values. For helping the nucleation, we run at $T = 500$ K for 2 ns using a Gilbert damping parameters $\alpha = 0.01$ at the positive fields. In the case of DG where the nucleation in the strong PMA layers could not always be achieved by solely playing with the temperature or the magnetic

noise, the nucleation was artificially induced by implementing one or two small skyrmions (radius of 15 nm) in the strong PMA layers and letting relax at different fields chosen to be coherent with the $R_{xy}$ measurements and the hysteresis measurements. The various components of the magnetization were then extracted and weighted if needed to compare with the transport measurements. Since the SMR is mostly an interfacial effect, the magnetization of each layer has not been normalized by its associated Co thickness, contrarily to the AHE and AMR which are more intertwined with the Co layer thickness.

In most of the simulations, Bloch lines are present due to the noisy initialization even though they are not energetically favorable. To reduce their number, the magnetization tensor was subsampled by a factor 4 and ran for 0.5 ns in Mumax3. Then, it was resized to its original size and finally relaxed to a stable state which was very similar to the original one, minus most of the Bloch lines.

The output of the micromagnetic simulations have been represented using the software Voxler4 after taking out the vacuum layers and using a multiplicative factor in the vertical direction of 3.3.

## Data availability
The data that support the findings of this study are available from the corresponding author upon reasonable request.

## Code availability
The code that supports the findings of this study is available from the corresponding author upon reasonable request.

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

## Acknowledgements

The authors are grateful to Yanis Sassi for advice on experimental techniques and simulations. financial supports from FLAG-ERA SographMEM (ANR-15-GRFL-0005), from ANR under the grant ANR-17-CE24-0025 (TOPSKY) and ANR-20-CE42-0012-01(MEDYNA) and as part of the 'Investissements d'Avenir' program SPiCY (ANR-10-LABX-0035).

## Author contributions

M.G. performed the experimental studies, the simulations and the analysis with the help of F.G. and S.C. for the growth, A.V., N.R., and K.B. for the MFM, N.R. for the micromagnetic calculations, N.R., V.C., and A.F. for the transport measurements. M.G., V.C., and N.R. conceived the experiments and wrote the manuscript.

## Competing interests

The authors declare no competing interests.
