## [Peer Review File · Nature Communications]

Three-dimensional skyrmionic cocoons in magnetic multilayersREVIEWER COMMENTS

Reviewer #1 (Remarks to the Author):

This publication describes a new type of 3D spin texture, the skymionic cocoon, which can be stabilised in magnetic multilayers. Interest in 3D spin textures has grown considerably over the past ten years and I find this article an interesting contribution. Overall, I find the work to have been carried out with a great level of rigour resulting in a well written article that is topical and suitable for the broad readership of Nature Communications.

I have only two comments which I hope will help to improve the manuscript:

1. Figure 1: I find that this portrays a lot of essential information for the reader and yet some of the scale bars, legends and arrow maps are not readable unless zooming to 200%. It may be best for this to be presented as a single column figure, or even separate figures allowing sub-sections to be expanded. This is really essential so the reader can understand the spin texture of the cocoons.

2. The authors only briefly (in the supplementary information), discuss the topological properties of these new spin textures. This is probably one of the most interesting aspects and probably requires some attention in the manuscript. Understandably, at this point only the 2D winding number has been calculated for each layer but I think even this short discussion would be useful in the main manuscript.

Reviewer #2 (Remarks to the Author):

In this work, the authors report the appearance of a skymionic localized structure in magnetic heterostructures formed by many layers, where thickness gradients are created as confinement potentials, with thickness values close to the spin reorientation transition. The paper is original and shows how the modulation of thicknesses in combination with the competition of magnetic energies in magnetic multilayers can potentially lead to 3D spin textures, of interest for future 3D spintronic technologies. However, I cannot currently recommend it for publication in Nature Communications due to the following reasons:

-The topological nature of skymionic cocoons is not studied with enough rigour in the paper, and this is a crucial point, given that the main message of the paper is the emergence of this new type of spin texture. Are these structures topologically trivial/nontrivial? A study using a 3D winding number formalism (see e.g. Nature Comm. 10, 593 (2019)) might be more appropriate than the current study included in supplementary using 2D skyrmion numbers.

-From a spintronics perspective, it is currently unclear what advantage this type of textures could potentially have with respect to for example isolated skyrmions located at different layers. The authors explain that it could be used to encode multiple states along the vertical position. Is there any interesting benefit of forming skymionic cocoons at different heights in the multi-layered system? Can the direct interaction between them or the engineered potential formed be exploited for 3D functionality?

-The properties of heterostructures with a large number of layers may degrade for the upper layers due to the propagation of structural defects, see e.g. Spin 3, 1340013 (2013). X-ray reflectometry or transmission electron microscopy measurements showing what is the quality of the layers as they are higher up the stack is a key aspect currently missing, since the degradation of properties could affect substantially the interpretation of the results.

In case that these points can be successfully answered, I have some questions and

comments:

- In equation 3, why is the planar Hall effect contribution discarded? The R_{xx} signals are systematically studied. However, R_{xy} is only shown for out-of-plane fields: why is this the case? It would help to validate the interpretation of the results by showing a consistent analysis of both R_{xx} and R_{xy} .
- The comparison of experiments and simulations is done for states following a different magnetic history: typically after saturation along one field direction for simulations, and after demagnetization for the experiments. Why is this a valid approach?
- Why are the magnetoelectrical measurements focused in the double-gradient sample, and those for the single-gradient sample are not shown? Is there also a good agreement in this second case?
- Given that the MFM contrast is such a vital tool to distinguish between the different phases in the system, can the authors include quantitative criteria to distinguish between the different textures depending on their structure and where they are located within the multilayer? Experiments and simulations were taken with tip-sample distance equal to 10 and 30 nm, respectively, so there is not a perfect quantitative agreement in between both.
- The references and initial discussion of the paper are overall excellent. However, the paper would benefit from including as well some previous works where the emergence of magnetic textures with complex profile along the vertical direction has been previously studied: For example, in the context of the spin-flop transition (Phys. Rev. Lett. 89, 127203 (2003)), by tuning boundary conditions (Phys. Rev. Lett. 112, 017206 (2013)), or by tuning thicknesses and RKKY coupling (Nature 493, 647 (2013)).

Reviewer #3 (Remarks to the Author):

In this manuscript magnetic multilayers of Al/Co/Pt are fabricated with varying Co thickness in order to modulate the PMA and DMI of the multilayers spatially over their thickness to form "Single Gradient" multilayers. Simulations show that these multilayers support skyrmion-tube/ chiral bobber/ toron like structures which extend partially through the thickness of the multilayers depending on the external magnetic field applied, these structures are dubbed skyrmion cocoons. Mumax3 Simulated MFM states as a function of external field are compared with experimental data and show qualitative agreement. A second sample is presented which is created by two SG multilayers separated by Al/Co/Pt multilayers with very thin Co, such that the PMA in the central layers is very large. Simulations show that skyrmion cocoons form in the top and bottom "SG" layers and can connect with each other depending on the external field value. MFM simulations are presented and agree qualitatively with experiment. Finally, both DG and SG samples are lithographically fabricated into micro structures for Hall bar measurements. Hall bar measurements show excellent agreement between simulation and theory for both the DG and SG structures.

This work is extremely significant to the field. The sample design, and general goal of the paper, are brilliant and pave the way to true three dimensional magnetic data storage by having different magnetic structures exist in separate layers of the material, while still allowing for structures that extend through the thickness of the material. I find the actual methodology and measurements (MFM+ hall bar measurements) scientifically sound. This work is detailed enough to be reproduced effectively.

Unfortunately there are several major problems (and some minor ones) that need to be

addressed before this work can be published.

1. MFM data could be from any sample with large DMI and interfacial PMA.

The mfm data, while matching simulation, shows nothing that would uniquely identify skyrmion cocoons. Specifically the MFM data qualitatively matches other multilayer systems with large DMI. For example (<https://www.nature.com/articles/s41467-018-08041-9>, <https://aip.scitation.org/doi/full/10.1063/1.5080713>, <https://www.nature.com/articles/nmat4934>

Considering that skyrmion cocoons are three dimensional structures, a purely two dimensional imaging technique does not discriminate between cocoons, double cocoons, or skyrmions. A depth sensitive measurement is needed. In order to make this measurement convincing, it should be able to clearly qualitatively (and quantitatively) discriminate between the different three dimensional structures the authors claim to have seen: cocoons that extend through the multilayer thickness, cocoons that partially extend through the thickness, and double cocoons where they are stacked on top of each other in different layers. As it currently stands, the MFM data does not support the three dimensional nature of the structures presented. Some papers where a depth sensitive measurement is used to identify a 3D structure by differentiating it from other similar structures:

<https://www.nature.com/articles/s41565-018-0093-3> and

<https://www.nature.com/articles/s41467-021-21846-5>

2. Lack of Uniqueness in Hall Bar measurements for different topological states.

This is a very similar critique as the last point. The Hall bar measurements show excellent quantitative agreement between simulation and experiment, but I'm not sure how they help distinguish between the different types of structures. Most notably in the SG sample there are never cocoons stacked on top of each other, but in the DG sample there is. Can these two states be distinguished using Hall bar measurements? If I was to give you a thin sample that only hosted skyrmions, how would one distinguish that from the more complex three dimensional structures? Within a single sample, is there a clear way of distinguishing states or identifying topological transitions?

I see no qualitative difference for hall bar measurements between SG and DG samples (comparing figures S6a,b and 4a+S6c). Given that lack of difference I do not see the hall bar measurements distinguishing between the single cocoon state, and the state where the cocoons stacked on top of each other.

3. Mfm simulation done with a lift of 30nm, but mfm experiment is done with a lift of 10nm

What does Mumax simulation data look like with 10nm lift? This is very easy to show/change.

4. It is not clear if skyrmion cocoons are new type of magnetic structure

I am not sufficiently convinced that these enclosed skyrmion cocoons are different from torons. The justification provided by the authors is "we only observe a texture resembling a radial vortex with a minute core having the same polarity as the background, induced by the dipolar field. Note the absence of Bloch points at the extremities of the cocoons that differentiate them from similarly shaped textures like the aforementioned dipole strings." To be clear: "A texture resembling a radial vortex with a minute core having the same polarity as the background" Is exactly what the top of a bloch point looks like in a toron. Fig 1d @ 275mt

looks like it does contain a block point. This could be improved with either a vertical cut in the x-z or y-z plane or a true three dimensional vector representation. For images of torons showing this type of bloch point see <https://aip.scitation.org/doi/pdf/10.1063/5.0033239> and <https://journals.aps.org/prb/pdf/10.1103/PhysRevB.98.174437>

I think the significance of the paper is not impacted by whether or not skyrmion cocoons are new structures, but it is essential to clarify.

5. Discussion of PMA variation.

In discussion of the DG sample the PMA variation due to the thickness variation of the multilayers is discussed. This needs to also be mentioned in the first part of the paper with the SG sample (only DMI variation is talked about in the first part of the paper).

6.Lack of true three dimensional representation of structures

The figures found in this paper are not up to the standards of high quality journals in this field; if presenting a three dimensional magnetic structure there should be a true three dimensional vector representation of such a structure. This should be trivial for the authors to fix as they have done their simulations in MuMax3 and can use existing software to create a true three dimensional representation, such as `muview2` (<https://grahamrow.github.io/Muview2/>). Specifically: Supplement Fig 2-d to have such a 3 dimensional vector representation. Figures like 2b, 3b are great. This could also help with critique #4.

The current state of the paper is not acceptable for publication and major revisions or additional experiments are needed to experimentally verify the three dimensional nature of the different structures presented. I hope that the authors are able to address most of these issues, as I do think that the ideas behind this paper are important and could be very relevant for future spintronic devices.

Reviewer #1

This publication describes a new type of 3D spin texture, the skyrmionic cocoon, which can be stabilised in magnetic multilayers. Interest in 3D spin textures has grown considerably over the past ten years and I find this article an interesting contribution. Overall, I find the work to have been carried out with a great level of rigour resulting in a well written article that is topical and suitable for the broad readership of Nature Communications.

We thank the Reviewer for her/his positive assessment of our work.

Q1.1 *I have only two comments which I hope will help to improve the manuscript:*

1. Figure 1: I find that this portrays a lot of essential information for the reader and yet some of the scale bars, legends and arrow maps are not readable unless zooming to 200%. It may be best for this to be presented as a single column figure, or even separate figures allowing sub-sections to be expanded. This is really essential so the reader can understand the spin texture of the cocoons.

In the revised version, we have tried to improve the layout of Fig. 1 as much as possible in order to highlight the essential information and evidence the spin textures of the cocoons.

Q1.2 *2. The authors only briefly (in the supplementary information), discuss the topological properties of these new spin textures. This is probably one of the most interesting aspects and probably requires some attention in the manuscript. Understandably, at this point only the 2D winding number has been calculated for each layer but I think even this short discussion would be useful in the main manuscript.*

We thank the Reviewer for this comment and agree that a more extended discussion about the topological properties of the textures is probably necessary, even though the topological nature of the 3D spin textures we observe is not their only interesting properties. As this question is also raised by Reviewer 2 (please see discussion below), we have included in the main text of the revised version, a short paragraph (given the lack of space) about the topological nature (or properties) of the cocoons. This is completed by a more extended presentation in the *Supplementary information* of how the different winding number formalisms i.e. 2D or 3D can be used in our case and what are the limits of validity of these approaches.

Reviewer #2

In this work, the authors report the appearance of a skyrmionic localized structure in magnetic heterostructures formed by many layers, where thickness gradients are created as confinement potentials, with thickness values close to the spin reorientation transition.

The paper is original and shows how the modulation of thicknesses in combination with the competition of magnetic energies in magnetic multilayers can potentially lead to 3D spin textures, of interest for future 3D spintronic technologies.

We are pleased that the Reviewer finds our work original and of interest.

Q2.1 However, I cannot currently recommend it for publication in Nature Communications due to the following reasons:

- The topological nature of skyrmionic cocoons is not studied with enough rigour in the paper, and this is a crucial point, given that the main message of the paper is the emergence of this new type of spin texture. Are these structures topologically trivial/nontrivial? A study using a 3D winding number formalism (see e.g. Nature Comm. 10, 593 (2019)) might be more appropriate than the current study included in supplementary using 2D skyrmion numbers.

We thank the Reviewer for this comment and explain after why we initially did not include it in the previous, as the intrinsic discrete nature of our material system might be not so appropriate in our case. In fact, the 3D skyrmion charge (not to be confused with the 2D magnetic skyrmions usually considered in the community) can be easily calculated. As discussed in the reference suggested by the Reviewer, as well as in [O. V. Pylypovskiy *et al*, Phys. Rev. B 85, 224401 (2012)] and [A. P. Malozemoff and J. C. Slonczewski, “Magnetic Domain Walls in Bubble Materials”, Academic Press, New York, 1979, p. 107], using the same notations, the 3D skyrmion charge is expressed by:

$$q = \frac{1}{4\pi} \int_S dA_i \epsilon_{ijk} \mathbf{m} \cdot \partial_j \mathbf{m} \times \partial_k \mathbf{m} = \pm 1 \quad ,$$

where the integral is taken over a surface S , enclosing the structure.

For a single enclosed Bloch point, q takes the value +1 or -1, and q can be zero for even number of Bloch points, like for the Bloch strings (a.k.a. dipole strings by some authors), also called torons or globules.

In our case, when the cocoon is fully buried inside the multilayer, it is obvious that the magnetization on a contour is uniform (or quasi-uniform) without curvature, and therefore leads to $q = 0$ (Figure R1). On the contrary, when the cocoon “emerges” at the surface of the multilayer, with a (2D-like) skyrmion in the topmost (or bottommost) magnetic layer, then q is entirely determined by the 2D skyrmion number of this last layer (Figure R1b).

Figure R1: Calculation of the 3D topological charge, neglecting the non-continuity of the magnetization along the growth direction. a) Selection of a cubic bounding box around a cocoon fully buried in the magnetic multilayer. A slice of the magnetization inside the box, indicated by the dashed line, is shown next to the bounding box. Obviously, there is no curvature of the magnetization at the surface of this bounding box, and therefore the topological number is zero. b) When the cocoon emerges to the surface of the magnetic multilayer, one side of the box displays a skyrmion texture, which topological charge is 1.

As already said, the lack of continuity (in the mathematical sense) of the magnetization in our “discrete” multilayers along the film normal, breaks the first hypothesis of these topological theorems. This is why we deliberately avoided initially to use the 3D winding numbers. However, we agree that the discussion is necessary. In the revised version, we include this description about the 3D charge number in the *Supplementary information*.

In complement, we point out that the layer-by-layer (2D) topology, which (approximatively) respects the very fundamental requirement of continuity, was already described in details in section 5 of the *Supplementary information*.

Q2.2 - *From a spintronics perspective, it is currently unclear what advantage this type of textures could potentially have with respect to for example isolated skyrmions located at different layers. The authors explain that it could be used to encode multiple states along the vertical position. Is there any interesting benefit of forming skyrmionic cocoons at different heights in the multi-layered system? Can the direct interaction between them or the engineered potential formed be exploited for 3D functionality?*

The architecture of a 3D device is the matter of a long paper in itself. The full exploitation of the potentialities of these 3D textures (or others) for spintronics is probably the matter of many targeted studies that goes well beyond this paper.

In order to answer more precisely this comment, some conceptual cocoon-based devices could be imagined for example for multiple states memory purposes. Indeed, a DG structure would naturally allow to encode a 4-states memory thanks to its various objects (single cocoons in bottom SG or in top SG, coupled cocoons or columnar skyrmions). By increasing the number of vertical positions available, such possibilities increase quickly. Another type of device leveraging the 3D textures could be for exploiting the spin wave dynamics in such 3D landscape to generate new magnonic behaviours. Finally, we cannot describe other types of devices targeting some neuromorphic functionalities as we would like first to protect the IP on these concepts.

Q2.3 *The properties of heterostructures with a large number of layers may degrade for the upper layers due to the propagation of structural defects, see e.g. Spin 3, 1340013 (2013). X-ray reflectometry or transmission electron microscopy measurements showing what is the quality of the layers as they are higher up the stack is a key aspect currently missing, since the degradation of properties could affect substantially the interpretation of the results.*

The structural analysis is indeed a key point discussing these complex structures, as the perpendicular magnetic anisotropy (PMA), for instance, will drop if the Pt|Co interface degrades.

Transmission electron microscopy for such type of complex and thick non-periodic multilayers is not an easy measurement, and moreover not easily accessible for us. Instead, we believe that the magnetic properties of our multilayers can be inferred from the excellent agreement with the numerical simulations that thus represent a good indication that the magnetic properties, including the PMA, are not impacted noticeably. This might be the case because we use relatively thick Pt interlayer of 2 to 3 nm, which helps smoothing the interfaces.

In Figure R2, we compare calculations with the X-ray reflectivity measurement of the double-gradient structure presented in the main text. The measurement was obtained by standard θ - 2θ scans with a Cu K_α laboratory source. A first qualitative observation is that we see peaks up to $2\theta = 9^\circ$, which is already a sign of low roughness. We tried to fit the data, but due to the plethora of parameters, 62 in our model, and the relatively poor fit of the resulting optimization, parameters are not reliably extracted.

However, by changing parameters in a controlled way in the calculations, we obtain qualitative information about the expected changes for several cases. The calculations confirm the empirical knowledge, oscillations are nearly fully damped at $2\theta = 5^\circ$ for excessive roughness of 1 nm (Figure R2). We also tried to determine if localized roughness (e.g. in the top part of the multilayer) could give rise to a specific signature. Unfortunately, the calculations in Figure R2b indicate that such approach seems inapplicable, due to the sensitivity to minute change in other parameters (thickness and density).

Figure R2: X-ray reflectivity of the double-gradient sample from the manuscript (grey lines) compared to several calculated curves: (a) uniform roughness through the multilayer, (b) using 0.5 nm everywhere in the sample except in specific areas where it is increased to 1 nm.

Q2.4 In case that these points can be successfully answered, I have some questions and comments:

- In equation 3, why is the planar Hall effect contribution discarded? The R_{xx} signals are systematically studied. However, R_{xy} is only shown for out-of-plane fields: why is this the case? It would help to validate the interpretation of the results by showing a consistent analysis of both R_{xx} and R_{xy} .

In such magnetic multilayers, the planar Hall effect (PHE) displays a sinusoidal behaviour when rotating the magnetic field in plane. Setting $\phi = 0$ along the x axis, that is the direction of the applied current, the maxima of the PHE are found at $\phi = \pi/4 + n\pi/2, n \in \mathbb{N}$ and importantly, PHE cancels along $\pm x$ and $\pm y$ (low and high conductivity axes). The corresponding measurement in the double gradient (DG) structure of the main text is shown in Fig. R3a along with the raw data of $R_{xy}(H_{\perp})$ to facilitate comparison. First, note that the amplitude of the PHE (18 m Ω) is about one fourth of the total amplitude of R_{xy} (81 m Ω). Moreover, this measurement corresponds to a saturated state, meaning that all the spins contribute equally to the PHE. However, when considering the R_{xy} measurement with an OOP field, only those present in the domain walls (the in-plane component) of the various textures contribute, and this greatly reduces its amplitude. Additionally, working under the assumption that the chirality and symmetry of the objects are well defined based on our simulations, the contribution for a Néel (or Bloch) *closed* domain wall (DW) should cancel out (see Figure R3). One could argue that this circular symmetry is not always preserved in the layers with strong PMA but due to their strong anisotropy, the associated DW width remains small which again correlates to a very low PHE contribution. As a sanity check, the average m_x and m_y have been extracted over the corresponding field range and the highest value was found to be lower than 5×10^{-3} over the whole simulation space, implying that the PHE should indeed be negligible.

We can easily generalize the previous reasoning to magnetic fields with a finite in-plane component ($\theta \neq 0^\circ$). Indeed, as it lies in the (yz) plane, locally, the spins will be more tilted along the y axis but their PHE contributions will still cancel each other so that the PHE remains neglectable. Even at saturation, the magnetization lies along a direction for which the PHE is null. Note that this conclusion would not stand for $\phi \neq n\pi/2$.

Figure R3: Study of the anomalous and planar Hall effects in the double gradient structure considered in the main text. a) Experimental raw data of $R_{xy}(H_{\perp})$ at $\theta = 90^\circ$, mostly related to anomalous Hall effect (AHE), and $R_{xy}(\phi)$ under a saturating field plotted with the same scale, displaying a clear planar Hall effect (PHE). The red line marks the half amplitude of each curve, *i.e.* the expected zero. It has a finite value due to a small spurious R_{xx} component. b) Planar cut of a skyrmionic cocoon with a Néel domain wall in presence of an OOP magnetic field. The circled green annotation corresponds to the sign of the PHE contributions.

In the main text, R_{xy} is only shown for an out-of-plane magnetic field as a way to emphasize the correlation with the micromagnetic simulations and thus to highlight the associated phase diagram. The complementary measurements for different magnetic field orientations can be found in the *Supplementary information* (Fig. S6) which shows the same quantitative agreement between the experimental and the numerical study.

Q2.5 - The comparison of experiments and simulations is done for states following a different magnetic history: typically, after saturation along one field direction for simulations, and after demagnetization for the experiments. Why is this a valid approach?

In micromagnetic simulations using *Mumax3*, the demagnetization process cannot be mimicked in a reasonable time so as a substitute, a saturation along the adequate axis is used. Additionally, a strong random noise is also introduced simultaneously which allows to artificially reproduce the experimental demagnetization.

Q2.6 - Why are the magnetoelectrical measurements focused in the double-gradient sample, and those for the single-gradient sample are not shown? Is there also a good agreement in this second case?

A similar study for the single-gradient structure was presented in the *Supplementary information* (Fig. S6) which also puts forward an excellent agreement between the simulations and the measurements. Note that some small discrepancies are visible: they mainly originate from the notorious difficulty to numerically nucleate textures, which often shifts the nucleation fields.

Q2.7 - Given that the MFM contrast is such a vital tool to distinguish between the different phases in the system, can the authors include quantitative criteria to distinguish between the different textures depending on their structure and where they are located within the multilayer? Experiments and simulations were taken with tip-sample distance equal to 10 and 30 nm, respectively, so there is not a perfect quantitative agreement in between both.

It is important to consider that MFM is mostly surface sensitive. However, in the case of double gradient structures, our multilayers can reach large thickness (close to 250 nm) so that the rapid decay of the stray field makes it impossible to actually pick up signal coming from the bottom layers. This suggests that the only objects that we are able to detect by MFM are (i) the skyrmionic cocoons located in the top gradient layers and (ii) the columnar textures extending over the full thickness. Between these two textures, the difference in contrast is sufficiently striking that they are easily identifiable (see Fig. 3 in the main text). A sound assumption concerning the bottom layers, supported by the micromagnetic simulations is that, given the symmetry of the stack and the need to minimize dipolar interaction, in most cases, cocoons will be present as well and aligned with the ones of the top gradient.

Regarding the MFM measurements on the single gradient sample in which a transition from skyrmion tubes to cocoons depending on the external field is observed, the decrease of the MFM phase contrasts with the field indicates that the textures are becoming more and more buried. To obtain more quantitative features, such as the actual depth in the vertical direction, we have to rely on the micromagnetic simulations which are in excellent agreement with the experimental characterization.

The remark addressing the difference of lift height between the experiments and the simulations is well founded. We address it in more details in the answer to the Reviewer 3.

Q2.8 - *The references and initial discussion of the paper are overall excellent. However, the paper would benefit from including as well some previous works where the emergence of magnetic textures with complex profile along the vertical direction has been previously studied: For example, in the context of the spin-flop transition (Phys. Rev. Lett. 89, 127203 (2003)), by tuning boundary conditions (Phys. Rev. Lett. 112, 017206 (2013)), or by tuning thicknesses and RKKY coupling (Nature 493, 647 (2013)).*

Those references put forward complex spin textures which indeed relates to our research but in their case, the system is confined in a single dimension. Indeed, the tuning of the thicknesses and RKKY coupling allows the manipulation of a soliton along the vertical axis which certainly yields a complex vertical evolution, but it remains a one-dimensional approach as the soliton can only move up or down. The same remark can be made about the stabilization of complex spin helices in one-dimensional systems by tuning the boundary conditions.

Reviewer #3

In this manuscript magnetic multilayers of Al/Co/Pt are fabricated with varying Co thickness in order to modulate the PMA and DMI of the multilayers spatially over their thickness to form "Single Gradient" multilayers. Simulations show that these multilayers support skyrmion-tube/chiral bobber/toron like structures which extend partially through the thickness of the multilayers depending on the external magnetic field applied, these structures are dubbed skyrmion cocoons. Mumax3 Simulated MFM states as a function of external field are compared with experimental data and show qualitative agreement.

A second sample is presented which is created by two SG multilayers separated by Al/Co/Pt multilayers with very thin Co, such that the PMA in the central layers is very large. Simulations show that skyrmion cocoons form in the top and bottom "SG" layers and can connect with each other depending on the external field value. MFM simulations are presented and agree qualitatively with experiment.

Finally, both DG and SG samples are lithographically fabricated into micro structures for Hall bar measurements. Hall bar measurements show excellent agreement between simulation and theory for both the DG and SG structures.

This work is extremely significant to the field. The sample design, and general goal of the paper, are brilliant and pave the way to true three-dimensional magnetic data storage by having different magnetic structures exist in separate layers of the material, while still allowing for structures that extend through the thickness of the material. I find the actual methodology and measurements (MFM+ hall bar measurements) scientifically sound. This work is detailed enough to be reproduced effectively.

We thank the Reviewer for his greatly positive outlook on our study.

Q3.1 *Unfortunately there are several major problems (and some minor ones) that need to be addressed before this work can be published.*

1. MFM data could be from any sample with large DMI and interfacial PMA.

The mfm data, while matching simulation, shows nothing that would uniquely identify skyrmion cocoons. Specifically the MFM data qualitatively matches other multilayer systems with large DMI. For example (<https://www.nature.com/articles/s41467-018-08041-9>, <https://aip.scitation.org/doi/full/10.1063/1.5080713>, <https://www.nature.com/articles/nmat4934>

Considering that skyrmion cocoons are three dimensional structures, a purely two dimensional imaging technique does not discriminate between cocoons, double cocoons, or skyrmions. A depth sensitive measurement is needed. In order to make this measurement convincing, it should be able to clearly qualitatively (and quantitatively) discriminate between the different three dimensional structures the authors claim to have seen: cocoons that extend through the multilayer thickness, cocoons that partially extend through the thickness, and double cocoons where they are stacked on top of each other in different layers. As it currently stands, the MFM data does not support the three dimensional nature of the structures presented. Some papers where a depth sensitive measurement is used to identify a 3D structure by differentiating it from other similar structures: <https://www.nature.com/articles/s41565-018-0093-3> and <https://www.nature.com/articles/s41467-021-21846-5>

In the references given by the Reviewer, most of the images show a phase distribution clearly centred around a single contrast (see Figure R4) and are usually associated with the 'standard' 2D-tubular skyrmions or worms that are invariant over the thickness. The measurement on the double gradient is strikingly different with indisputable two different contrasts that are visible (see Figure R4). This is directly linked to the stray field, thus it is a direct signature that different magnetic objects are coexisting.

Note that in some rare cases in the mentioned references, such as the bottom right images of Figure R4 different contrasts might be perceived which indicates that they are probably some strong variations in the vertical extension of the measured objects. We believe that this is likely due to the application of a perpendicular magnetic field that confines the textures. However, in our cases, the behaviour is different because different MFM contrasts can be detected over a large range of field, including at remanence, which is a key difference.

Figure R4: MFM comparison. a) Phase maps for the double gradient of the main text at two different fields, using a similar colour scale than in the references. b) Typical MFM images extracted from the aforementioned references (on the left, (d) and (f) from Fig. 3 of *Applied Physics Letters* 114.7 (2019): 072401, on the top right, g, h, i from Fig. 2 of *Nature materials* 16.9 (2017): 898-904, and bottom right c,d,e from Fig. 1 in *Nature communications* 10.1 (2019): 1-7.

Thus, the vastly different contrasts observed in MFM can easily be assigned to different magnetic objects with distinct vertical evolution. To resolve more precisely the separate structures visible in MFM, we rely on the micromagnetic simulations. More advanced experimental techniques like 3D tomography or holography with extended reference by autocorrelation linear differential operator (HERALDO) could be performed additionally but they are beyond the scope of this paper. In fact, we do have submitted proposals for beamtime on different synchrotron beamlines to try obtaining thickness-resolved measurements.

However, even without results using these complex 3D reconstruction techniques, we believe that the combination of MFM, magneto transport measurement and micromagnetic simulations allow to firmly confirm the existence of objects with different vertical extension.

For the identification of the various contrasts (columnar, single cocoon, paired cocoons), we refer to the answer given to Reviewer 2, Q2.7.

Regarding the SG samples, in which the evolution of the contrasts is less defined, we indeed solely rely on the corresponding micromagnetic simulations to study the vertical positioning of the spin textures but the excellent correlation with the magneto-transport measurements supports their veracity.

At last, we notice that even in the last reference proposed by the Reviewer for depth sensitive measurements, the authors insist that their “data do not constitute a direct and full 3D reconstruction of the spin texture with nanometre spatial resolution, which would be needed to claim [the stabilization of hopfions] without ambiguity”, showing the difficulty to resolve unequivocally a true 3D magnetization distribution.

Q3.2 2. Lack of Uniqueness in Hall Bar measurements for different topological states.

This is a very similar critique as the last point. The Hall bar measurements show excellent quantitative agreement between simulation and experiment, but I’m not sure how they help distinguish between the different types of structures. Most notably in the SG sample there are never cocoons stacked on top of each other, but in the DG sample there is. Can these two states be distinguished using Hall bar measurements? If I was to give you a thin sample that only hosted skyrmions, how would one distinguish that from the more complex three-dimensional structures? Within a single sample, is there a clear way of distinguishing states or identifying topological transitions?

I see no qualitative difference for hall bar measurements between SG and DG samples (comparing figures S6a,b and 4a+S6c). Given that lack of difference I do not see the hall bar measurements distinguishing between the single cocoon state, and the state where the cocoons stacked on top of each other.

We do not agree with the Reviewer on this point. Even if transverse resistance measurements appear to be qualitatively similar between SG and DG sample, they indeed differ *quantitatively*. However, as

clearly mentioned in the text, our confidence in the identification of the different textures comes principally from the comparison between the Hall measurements and the simulations.

To be a bit more precise in the argument, in the case of DG sample, there is no doubt that paired cocoons are present, thanks to the “magnetic” separation between the two gradient zone provided by the layers having a large PMA. The questioning of the Reviewer concerns more the differentiation between single cocoons or paired cocoons. This can be picked up by the amplitude of the magnetoresistance. A rule of thumb is that for two aligned cocoons the associated changes in the resistance will be close to being twice as large as for a single cocoon. This simple expectation is indeed confirmed by the simulations which usually shows paired cocoons and fit remarkably the magnetoresistance (see Fig. 4 of the main manuscript).

Moreover, as we have shown in *Supplementary information* Fig. S7, even a small modification of the z-position of the textures creates a noticeable difference in the resulting magnetoresistance. In order to further convince the Reviewer, we have tried in the simulations to delete the cocoons of the top layers while conserving the ones in the bottom layers. The result is that, for example at 325 mT with an OOP field (the relaxed states represented on Fig. 4 of the main text), $m_z = 0.848$ with cocoons in both gradients and 0.944 with only cocoons in the bottom one which corresponds to a difference of 5% in the signal. This corresponds to a clear experimental shift in the resistance of about 4 m Ω (*i.e.* about 25 times the noise level in the presented curve). Similarly, for R_{xx} , the difference between a single cocoon in one of the 2 gradients and 2 cocoons shall result in a 3 m Ω difference, that is again largely above the noise. Thus, we would expect a clearly different electrical signal in the case of having single cocoons in the DG structure. Those calculations have been used to update Fig. S7 to highlight the resistance difference between the two aforementioned states.

However, once again, we do not claim that we can resolve perfectly the 3D profile but that through the combined electrical measurements and simulations, the different states can be distinguished. Moreover, this approach is really much simpler (and adapted for potential future cocoon-based devices!) than any of the existing 3D-resolved imaging techniques.

Q3.3 3. *Mfm simulation done with a lift of 30nm, but mfm experiment is done with a lift of 10nm*

What does Mumax simulation data look like with 10nm lift? This is very easy to show/change.

We thank the Reviewer for this comment. In fact, the MFM data are acquired with the indicated lift height but the Reviewer should realize that the actual mean distance to the sample is higher than 10 nm (see Figure R5). Indeed, during the first pass which records the topography of the surface under consideration, the cantilever is already away for the surface given by about half the setpoint oscillation amplitude. The second pass, which measures the phase maps, is done at a given lift height, here 10 nm, indicating that the mean lift height is greater than the nominal one by this amount. However, we are not only changing the mean height above the surface but also the amplitude of oscillation by lowering the excitation ac voltage for the second pass in order to minimize the perturbation induced by the magnetic tip. In our previous work, a thorough study was led to determine the effective lift height (*Supplementary information of Nature materials* 19 (1), 34-42 (2020)) as reproduced in Figure R5. This yielded oscillation window ranging typically from 50 to 200 nm even for a lift height nominally equal to zero.

The story does not end at this point. One also needs to consider that the tip follows a sinusoidal behaviour and, therefore, it tends to spend more time near its extremal positions, *i.e.*, the closest and farthest distance away from the surface. Obviously, the strongest magnetic gradients are located near the surface which implies that most of the information will be picked up as the tip is in the lowest part of the oscillation. A weight function can thus be determined as a function of the effective height of the tip which display the expected decay (Figure R5) with a maximum near 50 nm for the tip under consideration. This defines an effective lift height. We added details in the methods of the main text, as “10 nm lift height” does not fully describe the experimental conditions.

Figure R5: Study of the lift height. a) MFM principle highlighting the definition of the lift height. b) Effective light height and oscillations amplitude for typical tips. Plot of the weight function of the information recorded during a given tip oscillation. Extracted from the Supplementary information of W. Legrand et al. *Nature materials* 19 (1) 34-42 (2020).

Beyond all these considerations, as shown in the Figure R6, a lift height of 50 nm does not change perceptibly the textures as simulated with a 30-nm lift height. The vertical variation of the stray field gradient in our multilayers is low enough, so that the simulated MFM signal is rather insensitive to the “lift height” parameter in *MuMax3*.

Figure R6: Comparison of the simulated MFM signal using a lift height of 30 and 50 nm.

Taking into account all these considerations, we updated the MFM image of the main text to display the *Mumax3* simulated MFM signal using an effective lift height of 50 nm.

Q3.4 4. *It is not clear if skyrmion cocoons are new type of magnetic structure*
*I am not sufficiently convinced that these enclosed skyrmion cocoons are different from torons. The justification provided by the authors is “we only observe a texture resembling a radial vortex with a minute core having the same polarity as the background, induced by the dipolar field. Note the absence of Bloch points at the extremities of the cocoons that differentiate them from similarly shaped textures like the aforementioned dipole strings.” To be clear: “A texture resembling a radial vortex with a minute core having the same polarity as the background” is exactly what the top of a bloch point looks like in a toron. Fig 1d @ 275mT looks like it does contain a block point. This could be improved with either a vertical cut in the x-z or y-z plane or a true three dimensional vector representation. For images of torons showing this type of bloch point see <https://aip.scitation.org/doi/pdf/10.1063/5.0033239> and <https://journals.aps.org/prb/pdf/10.1103/PhysRevB.98.174437>
 I think the significance of the paper is not impacted by whether or not skyrmion cocoons are new structures, but it is essential to clarify.*

The Reviewer is right: it is important to clarify if the texture that we call “cocoon” is a toron or a “new” texture deserving a new name. In fact, we thought about it deeply before introducing this new name.

The arrow scaling used for Fig. 1d brought maybe some ambiguities that, thanks to the Reviewer’s comment, we have corrected: the in-plane components are now properly scaled with the arrows’ length. The fact that the top layers at 275 mT in Fig. 1d are almost completely white means that the

magnetization lies almost completely in the z-direction. This is better seen in Figure R7 in which, as suggested by the Reviewer, we present an actual cut of m_z in different layers. It is hence clear that m_z in the layer n°13 (purple) is very close to 1. It means that the actual configuration does not correspond to the definition of a Bloch point (illustrated in Figure R7). It is also made clear with the vertical cut of the magnetization going through the cocoon centre (Figure R7) that no singularity is present: the magnetization rotates smoothly at the extremities of the cocoon. Moreover, we see that in layer 9 which marks the top of this particular cocoon, the magnetization configuration is mostly influenced by the dipolar interaction and no discontinuity is visible.

As for the references cited by the Reviewer, they describe a magnetic texture called a toron or in more recent papers a magnetic globule or dipole strings. Similar to the cocoons, it is characterized by a typical ellipsoidal shape. However, it possesses two opposite Bloch points of opposite topological charge, one at each extremity, which yields an overall topologically trivial nature (see also answer to Q2.1). It is also to be noticed that they have been observed in bulk chiral magnets and are thus defined in a continuous 3D volume (down to the atomic limit). In our case, the magnetization is intrinsically discretized in the z-direction due to the non-magnetic spacer used in the multilayers. The Bloch points (energetically costly) would not form *in* the magnetic layer, but would rather sit *in between* if the discrete layers of magnetization would be connected. Thus, the skyrmionic cocoons are, to our understanding, “topologically” different from the torons, and this is the reason why we decided to name them differently. We have chosen “cocoon” as it describes relatively well their shape but we might also have used the expression “*discrete torons*”.

The discussion related to Q2.1 and the present question leads to a clarification in the main text and a new section of the *Supplementary information* to clarify these aspects related to topology.

Figure R7: Study of the top layers for SG simulations of figure 1.d at 275 mT. a) Axial cut of m_z through the cocoons at various heights. b) Illustration of a Bloch Point (extracted from *Physical Review B* 101 (18), 184405 (2020)). c) Magnetization cuts evidencing the absence of Bloch Points near the cocoon extremities.

Q3.5 5. Discussion of PMA variation.

In discussion of the DG sample the PMA variation due to the thickness variation of the multilayers is discussed. This needs to also be mentioned in the first part of the paper with the SG sample (only DMI variation is talked about in the first part of the paper).

We thank the Reviewer for this comment and add in the revised version some extra information in the main text along with extra data in the *Supplementary information*, notably the evolution of the effective anisotropy with the Co thickness (see Fig. S2).

Q3.6 6. Lack of true three dimensional representation of structures

The figures found in this paper are not up to the standards of high quality journals in this field; if presenting a three dimensional magnetic structure there should be a true three dimensional vector representation of such a structure. This should be trivial for the authors to fix as they have done their simulations in MuMax3 and can use existing software to create a true three dimensional representation, such as mview2

(<https://grahamrow.github.io/Muview2/>). Specifically: Supplement Fig 2-d to have such a 3 dimensional vector representation. Figures like 2b, 3b are great. This could also help with critique #4.

We have tried to our best but even if the 3D representations are indeed trivial to produce, they are often very difficult to understand on a static illustration of small size, which is a reminiscent problem when dealing with 3D fields. In order to address this comment, we have tried several other options utilizing the full 3D vector field that are presented in Figure R8. In fact, we are not convinced by them as several drawbacks arise from such representations. First, to have correctly visualized the object, the field of view and the spins under consideration (up or down) must be restricted which makes it harder to visualize the vertical height of a given texture. Moreover, it is hard to access information inside the core of the object or to precisely understand the actual chirality of the spin texture, unless taking 2D cuts as we did in the main text. Our representations, using the iso-surfaces that the Reviewer praises and occasionally layer cuts to fully resolve the magnetization, thus appear to us more appropriate to evidence the evolution of the spin configuration.

Figure R8: 3D representation of a skyrmionic cocoons with vectorial field. Full view in 3D space and cuts in the (yz) plane exploring the various possible representations. The cocoons expand over 11 layers, the top 2 uniform layers are not shown in the left 3D view for simplicity.

The current state of the paper is not acceptable for publication and major revisions or additional experiments are needed to experimentally verify the three-dimensional nature of the different structures presented. I hope that the authors are able to address most of these issues, as I do think that the ideas behind this paper are important and could be very relevant for future spintronic devices.

We hope that the revision satisfies the problems arisen in the Review as we carefully consider all the associated insights and remarks.

REVIEWER COMMENTS

Reviewer #2 (Remarks to the Author):

The authors have successfully responded to all my concerns and questions. The additions to the manuscript after the first round of referees have improved the clarity and quality of the paper.
I therefore support its publication in Nature Communications in the current form.

Reviewer #3 (Remarks to the Author):

Please see the attached pdf.

Q3.1

Thank you very much for your answer, especially the comment on external magnetic field variation. I very much agree with the Authors on this now.

Q3.2

I very much Agree with the authors in the quantitative differences between the structures!

Q3.3

This is great! I will be citing the supplied article for future MFM work. Is this mentioned in the text of the paper? The authors should give this rationale for the mismatch between lift heights in simulation versus experiment in the paper.

Q3.4

- A) I don't believe discrete topological structures are distinguished from continuous ones, that is no one says "discrete skyrmions" for skyrmions in multilayers versus skyrmions in chiral crystals. I do think that skyrmion cocoons are torons, and chiral bobbbers. If the authors feel that they would like to call these structures skyrmion cocoons as they are a new magnetic system which can transform from from torons to chiral bobbbers, and exist in different layers, I think that is fine. As long as the connection to torons and chiral bobbbers is made clear.
- B) I am more convinced now that there is a bloch point. While Fig. R7b is a bloch point, there is more than one type of structure that is a bloch point. See Figure 1 of: <https://journals.aps.org/prb/abstract/10.1103/PhysRevB.85.224401>. The structures in Fig. 1 radially inverted are also bloch points. For example Fig.1 (a) of the referenced paper could also have all arrows point inward. The structure found in layer 9 is Fig 1(b) of the referenced paper radially inverted. I have drawn a red circle on the approximate center of the bloch point in layer 9 in Fig R7-edit.
- C) My response to Q2.1 is also related, and further confirms the necessary existence of a bloch point.

Ref Figure 1.

Fig. R7-edit

Q3.5. Great!

Q3.6 I really like the new figures in style of Fig. R7 c) and R8 (top right corner). I agree the muview figure is not helpful.

Q2.1.

From <https://doi.org/10.1080/00018732.2012.663070> equation 12

For topological structures defined by a winding number $d = d' + m + 1$

Where d is the dimension of the physical space, d' is the dimension of the defect, and m is the dimensionality of the membrane in order parameter space. A defect in physical space corresponds to a point where the order parameter vanishes, i.e. a bloch point for magnetic systems. To be clear the magnetization does not actually vanish at a certain point, but rather the center of the bloch point *exists in between spins* of the lattice.

For a classic 2D skyrmion $d=3$, $d'=0$, and, $m=2$. Bloch points are avoided.

For a 3D skyrmions $d=3$, $m=3$, and $d'=1$.

Therefore, any system with a three dimensional winding number has a bloch point. This is also explained clearly in the above reference.

The authors claim that in figure R1b, that the winding number of their three dimensional structure is 1. I think the authors have performed their topological charge calculation correctly. Therefore there is necessarily a bloch point somewhere in their structure, Fig R1b. It would most likely be at the bottom of the structure.

In figure R1a, they claim there is no net topological charge. This requires the generation of a bloch point with opposite winding number to the one that exists in R1b.

This is explained by the existence of bloch points near the top and bottom of the structure (in R1a) with opposite winding numbers. And hence these structures are torons, and chiral bobbbers.

Mathematically there are 2 options:

- A) There is no 3d winding number topological charge in Fig R1b and no bloch points.
- B) There is a 3d winding number topological charge in Fig R1b, and a bloch point.
Which requires Fig R1a to have two bloch points of opposite winding number so the net charge is 0. This is the most likely option. I frankly think it is very impressive that these are torons.

This was a very impressive R+R. I am thoroughly convinced that these structures have been verified experimentally, and I am very excited for this work to be published.

However, In order to be publishable the topological charge work either needs to be corrected, or removed completely.

Reviewer #2 (Remarks to the Author):

The authors have successfully responded to all my concerns and questions. The additions to the manuscript after the first round of referees have improved the clarity and quality of the paper.

I therefore support its publication in Nature Communications in the current form.

We thank the Reviewer for his/her positive evaluation of our work.

Reviewer #3 (Remarks to the Author):

Q3.1

Thank you very much for your answer, especially the comment on external magnetic field variation. I very much agree with the Authors on this now.

Q3.2

I very much Agree with the authors in the quantitative differences between the structures!

Q3.3

This is great! I will be citing the supplied article for future MFM work. Is this mentioned in the text of the paper? The authors should give this rationale for the mismatch between lift heights in simulation versus experiment in the paper.

Q3.4

A) I don't believe discrete topological structures are distinguished from continuous ones, that is no one says "discrete skyrmions" for skyrmions in multilayers versus skyrmions in chiral crystals. I do think that skyrmion cocoons are torons, and chiral bobbbers. If the authors feel that they would like to call these structures skyrmion cocoons as they are a new magnetic system which can transform from from torons to chiral bobbbers, and exist in different layers, I think that is fine. As long as the connection to torons and chiral bobbbers is made clear.

B) I am more convinced now that there is a bloch point. While Fig. R7b is a bloch point, there is more than one type of structure that is a bloch point. See Figure 1 of: <https://journals.aps.org/prb/abstract/10.1103/PhysRevB.85.224401>. The structures in Fig. 1 radially inverted are also bloch points. For example Fig.1 (a) of the referenced paper could also have all arrows point inward. The structure found in layer 9 is Fig 1(b) of the referenced paper radially inverted. I have drawn a red circle on the approximate center of the bloch point in layer 9 in Fig R7-edit.

C) My response to Q2.1 is also related, and further confirms the necessary existence of a bloch point.

Ref Figure 1.

Fig. R7-edit

Q3.5. Great!

Q3.6 I really like the new figures in style of Fig. R7 c) and R8 (top right corner). I agree the muview figure is not helpful.

Q2.1.

From <https://doi.org/10.1080/00018732.2012.663070> equation 12

For topological structures defined by a winding number $d = d' + m + 1$

Where d is the dimension of the physical space, d' is the dimension of the defect, and m is the dimensionality of the membrane in order parameter space. A defect in physical space corresponds to a point where the order parameter vanishes, i.e. a Bloch point for magnetic systems. To be clear the magnetization does not actually vanish at a certain point, but rather the center of the Bloch point exists in between spins of the lattice.

For a classic 2D skyrmion $d=3$, $d'=0$, and, $m=2$. Bloch points are avoided.

For a 3D skyrmions $d=3$, $m=3$, and $d'=1$.

Therefore, any system with a three dimensional winding number has a Bloch point. This is also explained clearly in the above reference.

The authors claim that in figure R1b, that the winding number of their three dimensional structure is 1. I think the authors have performed their topological charge calculation correctly. Therefore there is necessarily a Bloch point somewhere in their structure, Fig R1b. It would most likely be at the bottom of the structure. In figure R1a, they claim there is no net topological charge. This requires the generation of a Bloch point with opposite winding number to the one that exists in R1b.

This is explained by the existence of Bloch points near the top and bottom of the structure (in R1a) with opposite winding numbers. And hence these structures are torons, and chiral bobbers.

Mathematically there are 2 options:

A) There is no 3d winding number topological charge in Fig R1b and no Bloch points.

B) There is a 3d winding number topological charge in Fig R1b, and a Bloch point. Which requires Fig R1a to have two Bloch points of opposite winding number so the net charge is 0. This is the most likely option. I frankly think it is very impressive that these are torons.

This was a very impressive R+R. I am thoroughly convinced that these structures have been verified experimentally, and I am very excited for this work to be published. However, in order to be publishable the topological charge work either needs to be corrected, or removed completely.

Again, we thank the Reviewer for raising this interesting question and we believe that the question of topology and nomenclature brought up by the Referee are indeed strongly related. We concur with his/her definition about Bloch points (BP) and their relation with torons which is why we clarified in the main text by adding the term 'toron' to the 'dipole string' one.

We fully agree with all the of the Referee's comment, however as long as a continuous magnetic medium is considered. This is not the case in our magnetic multilayers that are discretized along the z-direction. Indeed, in our case, the interlayer distance between two ferromagnetic layers is typically 4.4 nm which is too large to allow any significant interlayer symmetric exchange interaction. In that case, the equilibrium is primarily resting on the dipolar interaction. In such case, an abrupt rotation of the magnetization from one layer to another is not prohibited (see Fig. SR1a). In our opinion, there is consequently no need to have a Bloch point that would ensure the smooth rotation of the magnetization. Moreover, for the sake of argument, even if one Bloch point was to exist, it would have to sit in between two consecutive magnetic layers, that is surrounded by non-magnetic atoms, hence not being really compatible with the definition of a Bloch (at least shown those in the Figure proposed by the Referee in his/her report).

Pushing the reasoning *ad absurdum*, we show in Fig. SR1c, that at some point, defining a Bloch point in between layers lacks sense (one could add as many Bloch points as one would like). The limit up to which it is still meaningful to consider the existence of a Bloch point wasn't discussed yet, as far as we

know, but, to us, it seems necessary to keep a “strong enough” Heisenberg-like symmetric interaction in between the layers, the only term ensuring some continuity in the magnetization textures.

As an additional note, the Referee claims that a Bloch point is visible in the magnetization cut shown below. According to our previous argument, due to the discontinuity along the z-direction it should not correspond to one. Moreover, it actually does not correspond to the extremity of a cocoon (as expected for torons) as shown with the cut of the layer below (see Fig. SR1b) and is rather a Bloch line in the “wall” of the 2D skyrmion in the layer 9.

All this reasoning made that we tentatively introduced the name ‘skyrmionic cocoons’ in an attempt to put forward this difference of the textures that we observe with already existing magnetic textures. Note that the Reviewer 3 is also right about the fact that the ‘skyrmions’ in discrete multilayers, that have been investigated since a decade, might also have discontinuous textures along the out-of-plane direction and therefore could be, or even should be called discretized ‘skyrmion tubes’ in order to better described their actual textures. In fact, in some of our recent publications (see our Ref. 5, W. Legrand et al., “Hybrid chiral domain walls and skyrmions in magnetic multilayers” *Sci. Adv.* 4, eaat0415 (2018)), we even demonstrated experimentally and numerically that under certain conditions, the actual effective chirality of these skyrmion tubes is changing along the z-direction.

Figure SR1. a) Simulated pair of stacked layers highlighting the absence of smooth rotation of the magnetization between two consecutive layers: the cocoons wall can rotate by more than 90° between layer 6 and 7. This is due to the interlayer spacing nullifying the exchange interaction so that the equilibrium is mainly governed by the dipolar interaction, rendering unessential the existence of BP. b) Zoom on the top of a cocoon in the top two layers with the green circle indicating the potential presence of a BP. c) Increasing the separation of the layers above some threshold inevitably leads to the loss of Bloch points (BP).

REVIEWERS' COMMENTS

Reviewer #3 (Remarks to the Author):

Mathematically, there are still only two options:

“A) There is no 3d winding number topological charge in Fig R1b and no Bloch points.
B) There is a 3d winding number topological charge in Fig R1b, and a Bloch point. Which requires Fig R1a to have two Bloch points of opposite winding number so the net charge is 0.”

Any argument that invalidates the existence of a Bloch point will also invalidate a 3D winding number.

In this specific case, the authors argue that the system is too discrete in the vertical direction to have a Bloch point. If the system is that discrete in the vertical direction, then you can't have a 3D winding number; one of the first approximations made when calculating winding numbers/ topological charge is that the system is continuous.

I really think option A) or B) are equally fine. Based on comments from the authors I think they would prefer to make the argument that their system is discrete in the vertical direction and therefore it doesn't have a 3d winding number topological charge, it doesn't have Bloch points, and it is structurally distinct from torons and chiral bobsers.

Otherwise this is very publishable. I don't think another round of R+R is needed if the authors can work out this issue with the editor. Even if this issue can't be resolved to my satisfaction (i.e. choose A or B), with the added explicit connection to torons a reader could work out that in a continuous system Bloch points are needed.

ANSWER TO REVIEWER 3

Reviewer #3 (Remarks to the Author):

Mathematically, there are still only two options:

“A) There is no 3d winding number topological charge in Fig R1b and no bloch points.

B) There is a 3d winding number topological charge in Fig R1b, and a bloch point. Which requires Fib R1a to have two bloch points of opposite winding number so the net charge is 0.”

Any argument that invalidates the existence of a Bloch point will also invalidate a 3D winding number. In this specific case, the authors argue that the system is too discrete in the vertical direction to have a bloch point. If the system is that discrete in the vertical direction, then you cant have a 3D winding number; one of the first approximation made when calculating winding numbers/ topological charge is that the system is continuous.

I really think option A) or B) are equally fine. Based on comments from the authors I think they would prefer to make the argument that their system is discrete in the vertical direction and therefore it doesn't have a 3d winding number topological charge, it doesn't have bloch points, and it is structurally distinct from torons and chiral bobbbers.

Otherwise this is very publishable. I don't think another round of R+R is needed if the authors can work out this issue with the editor. Even if this issue can't be resolved to my satisfaction (i.e. choose A or B), with the added explicit connection to torons a reader could work out that in a continuous system bloch points are needed.

We still agree with Reviewer 3: we think that there is no 3D winding number, i.e. option A. We took many precautions to make this point clear in the main text:

*“Alternatively, we can consider the 3D generalization of the topological charge (as defined in [30]) for which the integral is performed on a closed surface enclosing the magnetic object, **heeding that this definition is ill-defined as we study a non-continuous medium.** In the Supplementary Information (Section 5), we detail how the calculated topological charge **would** depend on the extension and position of the cocoon **in case the magnetization can be considered continuous.**”*